# HIV-1 accessory protein Vpr possesses a cryptic p300-dependent transcription-promoting activity that is blocked by histone deacetylases in CD4+ T cells

Catherine A. Lewis[1,2], David M. Margolis[1,2,3], Edward P. Browne[1,2,3]*

**1** Department of Microbiology and Immunology, University of North Carolina at Chapel Hill School of Medicine, Chapel Hill, North Carolina, United States of America, **2** University of North Carolina HIV Cure Center, University of North Carolina at Chapel Hill School of Medicine, Chapel Hill, North Carolina, United States of America **3** Department of Medicine, University of North Carolina at Chapel Hill School of Medicine, Chapel Hill, North Carolina, United States of America,

* epbrowne@email.unc.edu

## Abstract

Antiretroviral therapy (ART) has dramatically improved the clinical prognosis for people with HIV and prevents HIV transmission. However, ART does not cure HIV infection because of a persistent, latent viral reservoir in long-lived cells such as central memory CD4+ T ($T_{CM}$) cells. Eliminating or preventing reservoir formation will require a better understanding of HIV-1 latency establishment. We and others have recently shown that host cell factors such as histone deacetylases (HDACs) are critical cellular factors that allow HIV-1 entry into latency. Whether HDACs interact with specific viral factors to regulate latency establishment, however, is unknown. To examine the role of individual HIV-1 accessory proteins, we constructed a panel of HIV-1 reporter strains, each expressing a single HIV-1 accessory protein, and examined them in a primary CD4+ T-cell latency model. Interestingly, we found that the HDAC inhibitor (HDACi) vorinostat potently enhances the effect of the HIV-1 protein Vpr in promoting HIV expression in infected cells, suggesting that Vpr possesses a cryptic transcription-promoting activity that is restricted by HDACs. This activity was dependent on a p300-binding domain of Vpr and inhibited by a selective p300 histone acetyltransferase inhibitor. Interestingly, Vpr expression also resulted in a significant increase in the proportion of infected cells with a central memory ($T_{CM}$) phenotype. Furthermore, we observed that $T_{CM}$ cells were more resistant to Vpr-induced apoptosis/cell death than other CD4+ T-cell subtypes, indicating that Vpr expression during reservoir formation selects for latent proviruses in $T_{CM}$ cells. Overall, these findings suggest that Vpr plays an important role in shaping the latent reservoir and that HIV-1 latency results, in part, from an HDAC-mediated restriction of Vpr's transcription-promoting activity. Understanding how viral factors shape the latent

PLOS Pathogens

**Data availability statement:** Data are available on UNC Dataverse at https://doi.org/10.15139/S3/EYK07O.

**Funding:** This work was supported by the following grants from the National institutes of Health: NIAID #5-R01AI143381 (EPB) and NIAID #5-UM1AI164567 (DMM) (https://www.niaid.nih.gov/grants-contracts/opportunities). The funders had no role in study design, data collection and analysis, decision to publish, or preparation of the manuscript.

**Competing interests:** The authors have declared that no competing interests exist.

reservoir and how host and viral factors interact during HIV-1 latency establishment in CD4+ T cells will aid in the development of new latency-targeting therapies.

## Author summary

Although antiretroviral therapy is effective at treating HIV, a cure remains elusive. The primary obstacle to HIV cure is the presence of a long-lived reservoir of latently infected cells in which the virus persists despite therapy. Recent work has shown that a sizable fraction of this latent reservoir forms near the time that therapy is initiated, suggesting it may be possible to prevent some of the reservoir from forming. However, latency prevention will require a better understanding of how HIV enters latency, including how viral gene expression is silenced. We therefore sought to examine the role of the interaction between viral proteins and host factors in turning off viral gene expression and found that, whereas the HIV protein Vpr turns on viral gene expression, host histone deacetylases block this activity. Second, we observed that Vpr expression in infected cells leads to an increase in the relative proportion of central memory CD4+ T cells, a cell type that harbors latent virus. Our findings on the role of the viral protein Vpr in the silencing of viral gene expression and the persistence of certain memory cell types during infection will be important for developing new approaches to targeting latently infected cells.

## Introduction

Although combined antiretroviral therapy (ART) has significantly reduced human immunodeficiency virus 1 (HIV-1) transmission and has improved prognosis for people with HIV-1 (PWH), ART is still associated with comorbidities [1,2] and PWH still experience problems of access and stigma [3–5]. Furthermore, ART is not a cure for HIV infection. ART suppresses viremia by targeting actively replicating virus, but the virus persists as a latent reservoir of stably integrated proviruses and the rate of reservoir decay is too slow for the virus to be cleared during a PWH's lifetime [6–8]. Because of the HIV-1 reservoir, viral levels rapidly rebound if ART is interrupted, demonstrating the urgent need for a cure that either eradicates or permanently silences the latent reservoir.

The latent reservoir begins forming soon after initial infection, and infected cells can undergo clonal expansion to replenish the reservoir even during ART [9–12]. One major approach to HIV-1 cure, therefore, has been to develop latency-reversal agents (LRAs) that target host factors involved in HIV-1 transcriptional regulation, with the goal of reactivating viral gene expression to a sufficient extent that the host immune system can detect and kill infected cells. However, while many LRAs can reactivate detectable viral gene expression, latency reversal alone has not yet led to a significant reduction in the frequency of latent infection [13–18]. This lack of reservoir

PLOS Pathogens

reduction is likely due to latency being maintained by multiple levels of epigenetic and transcriptional repression, making it challenging to broadly reactivate the reservoir by modulating a single mechanism.

In recent years, several studies have provided evidence that a large fraction of the long-lived latent reservoir is formed or stabilized near the time of ART initiation [19,20]. This finding suggests that it may be possible to intervene during the initiation of ART and prevent a significant part of the reservoir from forming. Because repression of HIV-1 is established gradually in a step-wise manner [21,22], such an approach may be more effective than reversing latency in proviruses after several years of therapy. However, preventing latency will require an improved understanding of how viral latency is established. We previously reported that epigenetic reprogramming via histone deacetylase (HDAC) activity is a critical early step for entry into latency that licenses subsequent repressive histone methylation [21]. Furthermore, we showed that HDAC activity may also help to maintain T cells in a long-lived, stem cell memory-like state ($T_{SCM}$ cells), a memory subset in which the long-lived latent reservoir has been reported to persist [21,23].

In this study, we sought to understand how HIV-1 accessory proteins shape viral entry into latency, both alone and in combination with HDACs. We report that the viral accessory protein Vpr has a strong effect that appears to be more than additive on viral gene expression when combined with the class I HDAC inhibitor (HDACi) vorinostat, suggesting that Vpr possesses a cryptic activity in CD4+ T cells that counteracts latency establishment but is blocked by HDACs. Furthermore, this activity depends on a p300-binding domain of Vpr and histone acetyltransferase activity of p300. We also observed that Vpr has a striking effect on the phenotype of cells that enter the pool of latently infected cells by selecting for cells with a long-lived $T_{CM}$ phenotype. Our data suggest that Vpr expression during latency establishment plays an important role in shaping key characteristics of the latent reservoir.

## Results

### Vpr expression antagonizes HIV latency establishment in the presence of a histone deacetylase inhibitor (HDACi)

To investigate the role of HIV-1 accessory proteins in latency establishment, we used a primary CD4+ T-cell model of HIV latency that we have previously established. This model uses a reporter strain of HIV (HIV-GFP) that lacks expression of all viral proteins except Tat and Rev [21,24,25] but expresses a destabilized GFP gene and the surface marker Thy1.2. Expression of both GFP and Thy1.2 is driven by the endogenous viral LTR promoter, and the greater stability of the Thy1.2 protein allows us to identify cells that have recently downregulated HIV gene expression as GFP- cells within the Thy1.2+ population. In this model, activated CD4+ T cells are infected with HIV-GFP before being cultured for up to three weeks. Over this three-week period in this single-round infection system, viral gene expression progressively declines and a latently infected population with low to undetectable GFP expression (GFP- within Thy1.2+) emerges from the actively infected population (GFP- within Thy1.2+). To study the effect of individual HIV-1 proteins on latency initiation in this system, we generated single gene revertant clones of HIV-GFP for *Gag/Pol*, *Vif*, *Vpu*, *Vpr*, and *Nef*. We confirmed successful reversion of each gene and rescue of Gag/Pol, Vif, and Vpr protein production by sequencing of the plasmid and western blot of transfected 293T cells, respectively (S1A Fig). Expression of the two reporter genes, Thy1.2 and GFP, was confirmed in infected CD4+ T cells for all revertant viruses (S1D Fig). There was some variation in the percentage of GFP+ cells following infection with the different revertant viruses, possibly due to the differing biological activity of each rescued protein. For example, the abundance of cells infected with Vpr-expressing virus was typically lower at later timepoints, likely due to the known ability of Vpr to promote apoptosis. Of note, CD4 expression was downregulated in cells infected with Nef or Vpu revertant virus, confirming functional rescue of these HIV-1 proteins (S1B Fig).

We then examined the effect of each viral gene individually on establishment of HIV latency in CD4+ T cells. Recently activated CD4+ T cells were infected with infectious supernatant containing VSV-G-pseudotyped HIV-GFP as well as each of the revertant virus clones, and, over the following three weeks, flow cytometry was used to measure the percentage of productively infected cells (%GFP+ cells within the Thy1.2+ population) and the median fluorescence intensity (MFI) of

GFP expression within the GFP$^+$ population. When we examined the overall effect of the revertant viruses on the percentage of productively infected cells and GFP MFI within GFP$^+$ cells, we did not observe a large difference between the revertant viruses and the parental virus strain for Gag/Pol, Nef, Vif or Vpu (Fig 1D, 1F). Although previous studies have shown that Vpr can transactivate HIV-1 expression [26–31], we also did not observe a strong effect of Vpr on the percentage of productively infected cells in this model system and we only observed a statistically significant effect of Vpr on viral gene expression by GFP MFI on day 7 (q < 0.05 based on adjusted p-values by the FDR method of Benjamini and Hochberg; Figs 1D,1F, 2C). These results indicate that, individually, the HIV proteins Gag/Pol, Vif, Vpr, and Nef do not have a strong effect on HIV transcriptional downregulation or entry into latency in infected CD4$^+$ T cells.

We previously showed that, in this system, histone deacetylases (HDACs) play an important role in initiating transcriptional silencing of the HIV-1 promoter (long terminal repeat; LTR) during latency establishment [21]. We therefore sought to assess the relationship between individual viral proteins and HDACs by treating cells with the class I HDAC inhibitor (HDACi) vorinostat or vehicle control (DMSO) every 2–3 days from two days post-infection onwards. As we previously observed, vorinostat alone resulted in a large fraction of the infected cells failing to enter latency and remaining GFP$^+$ [21], whereas a substantial fraction of DMSO control-treated cells became GFP$^-$ by 21dpi (Fig 1C). Of note, although we did not see a significant change in the percentage of productively infected (GFP$^+$) cells following infection with any of the revertant viruses (Fig 1E), vorinostat exposure led to a significant increase (q < 0.01) in viral gene expression (GFP MFI within the GFP$^+$ cell gate) in cells infected with Vpr-expressing revertant compared with cells infected with parental virus HIV-GFP (Figs 1G, 2C). The combination of vorinostat treatment and Vpr expression led to strongly maintained viral gene expression over time, an effect that was not observed with either vorinostat or Vpr alone (Figs 1D,1G, 2C). Vorinostat is a pan-class I HDACi and inhibits HDAC1, 2, 3, 8 and also likely class II HDAC6. To identify the HDAC responsible for counteracting the pro-transcriptional activity of Vpr, we treated HIV-GFP or HIV-GFP-Vpr-infected cells with a selective HDAC1/2i, an HDAC3i, or vorinostat starting two days post-infection and every three days subsequently and assessed the effect on GFP MFI within GFP$^+$ cells after 14 days. Due to the similarity of HDAC1 and HDAC2, the HDAC1/2i potently targets both of these HDACs. The HDAC1/2i and HDAC3i were provided by Merck, and we have previously published their respective structures [21]. We found that HDAC1/2i had a strong positive effect on viral gene expression for the Vpr-expressing virus, similar to the effect of vorinostat, suggesting the importance of HDAC1 or HDAC2 in counteracting the effect of Vpr on viral gene expression (S2A Fig).

Interestingly, we found that later vorinostat treatment of HIV-GFP- or HIV-GFP-Vpr-infected cells at 14dpi, when the majority of the proviruses are in a latent state, did not sizably increase the fraction of productively infected cells or GFP MFI within productively infected cells (S2B, S2C Fig). Given that overall viral protein expression is relatively low at 14dpi, we speculate that the lack of combined effect of Vpr and vorinostat at 14dpi is due to insufficient Vpr or HDAC activity at this later timepoint. Overall, these data indicate that Vpr possesses an activity that can potently counteract downregulation of viral gene expression in recently infected CD4$^+$ T cells but is blocked by the action of HDACs.

## Vpr expression *in cis* or *in trans* in combination with an HDACi enhances HIV transcription

Because the HIV LTR is repressed by HDACs during latency and the *Vpr* gene of HIV is expressed in an LTR-dependent manner, we hypothesized that the strong combined effect of HDACis and Vpr expression might be due to upregulation of Vpr expression itself by HDACis. Thus, to determine whether Vpr in combination with HDACi enhances HIV expression when Vpr expression is LTR-independent, we used a co-infection model: Primary CD4$^+$ T cells were co-infected with HIV-GFP and an EF1$\alpha$-driven lentiviral vector expressing either mCherry alone or co-expressing Vpr and mCherry (mCherry and Vpr-mCherry, respectively) followed by vorinostat treatment beginning at 2–3dpi and monitored by flow cytometry for 21 days (Fig 2A, 2B). Similar to our results with cis Vpr expression (Fig 2C), co-infected cells (GFP$^+$mCherry$^+$) with trans Vpr expression and treated with vorinostat had higher viral gene expression than either vorinostat-treated cells co-infected with mCherry lentivirus or DMSO-treated cells co-infected with the Vpr-mCherry lentivirus, particularly on day 7 (q < 0.05

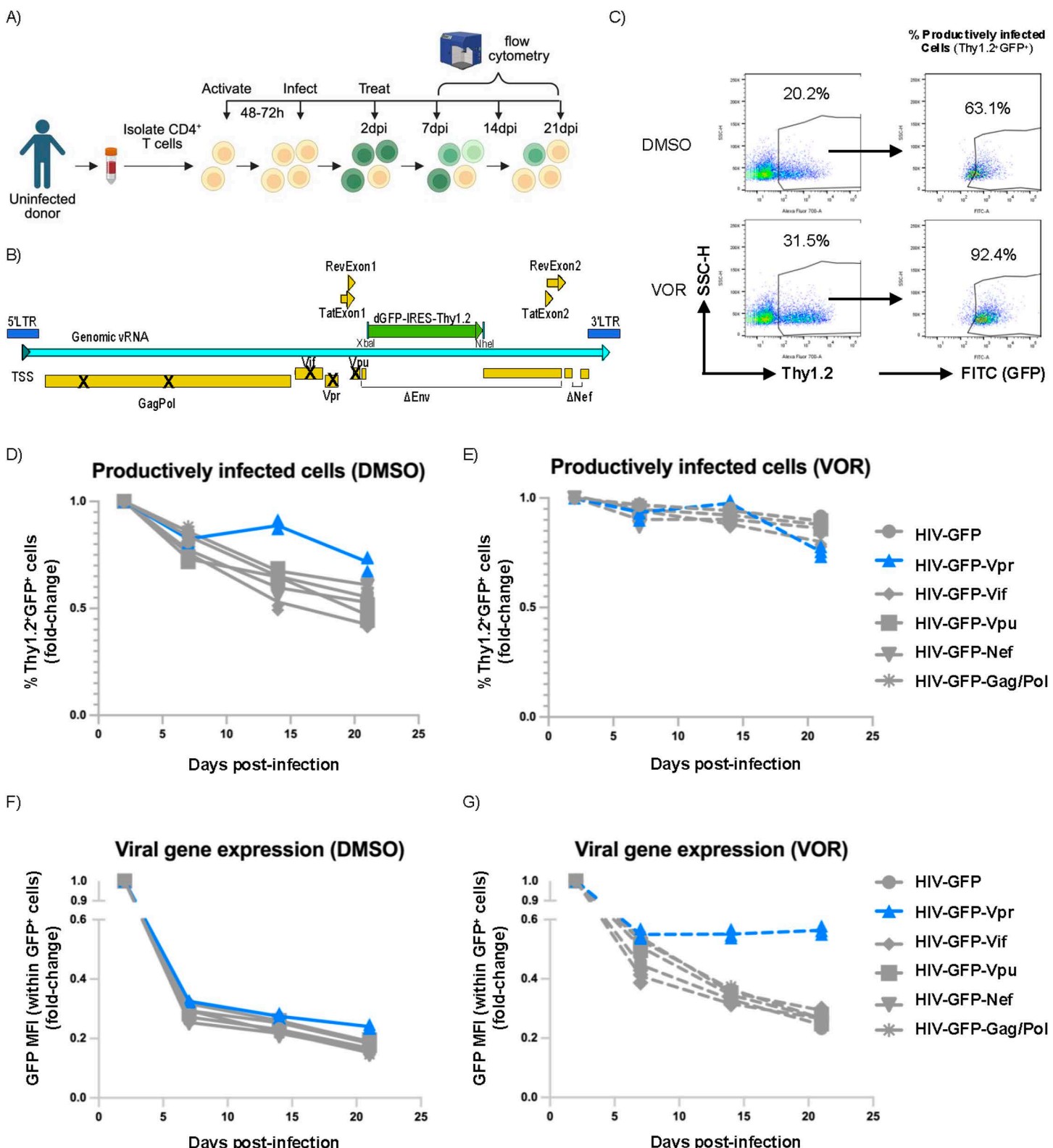

**Fig 1. Effect of single viral gene revertants on productive infection and viral gene expression.** A-B) Primary CD4+ T-cell model **A)** Overall workflow (created with Biorender.com). **B)** HIV-GFP-Thy1.2 construct. Bold Xs represent point mutations that result in premature stop codons. ΔEnv represents the portion of the Env-encoding region that has been replaced with a destabilized eGFP followed by an IRES-Thy1.2 cassette. ΔNef indicates a

truncation in the Nef-encoding region. TSS = transcription start site. **C)** Gating strategy for Thy1.2 and GFP (within Thy1.2$^+$ cells) and representative plot of median fluorescence intensity (MFI) within GFP$^+$ cells. Top plots are vehicle control (DMSO)-treated. Lower plots are vorinostat (VOR)-treated. D-E) %productively infected cells (Thy1.2$^+$GFP$^+$) following infection with revertant virus and **D)** DMSO or **E)** VOR treatment. F-G) GFP MFI within productively infected cells following **F)** DMSO or **G)** VOR treatment. n = 2 donors for all experiments.

by Benjamini and Hochberg procedure; Fig 2D). Together, these results indicate that HDACs block a transactivation activity of Vpr in CD4$^+$ T cells, whether Vpr is expressed in cis or in trans, and that the potent combined effect of Vpr and vorinostat is independent of the promoter driving Vpr expression.

Because viral gene expression is measured via levels of a reporter protein (GFP), these data could reflect an effect on viral transcription or also an effect on a post-transcriptional step such as RNA splicing, translation, or protein stability. To determine whether HIV-1 transcription levels are increased following Vpr expression and vorinostat treatment, at 14dpi, we sorted GFP$^+$ cells from vorinostat or DMSO-exposed cells (n = 2) that were infected with HIV-GFP-Vpr or HIV-GFP and measured Gag unspliced mRNA (Gag usRNA) by reverse-transcription qPCR. We observed a 13-fold increase in Gag usRNA with combined HIV-GFP-Vpr infection and vorinostat exposure compared with HIV-GFP infection with DMSO exposure. By contrast, we observed approximately 2.5-fold increases in Gag usRNA with either HIV-GFP-Vpr infection or vorinostat treatment alone (Fig 2E). We also confirmed this observation with lentiviral-driven Vpr expression in trans (Fig 2F). Overall, these data demonstrate that Vpr and vorinostat combine to significantly upregulate HIV expression and counteract latency establishment at the transcriptional level.

### Vorinostat treatment does not alter the subcellular localization of Vpr

Next, we hypothesized that vorinostat could modify or enhance Vpr's transcription-enhancing activity by affecting its subcellular localization. In previous studies, largely performed in cell lines, Vpr has primarily been found in the nucleus and nuclear envelope but may also be present in the cytoplasm and could affect HIV expression through its activity in either compartment [32–34]. For example, vorinostat could potentially promote increased nuclear import of Vpr, thereby allowing it to directly enhance HIV expression in the nucleus. To assess whether vorinostat treatment affects Vpr localization within infected cells, we fractionated vorinostat- or DMSO-treated cells that had been infected with either HIV-GFP or HIV-GFP-Vpr into chromatin, soluble nuclear, membrane, and cytosolic fractions and then examined the abundance of Vpr in each compartment by western blotting for Vpr. To confirm successful fractionation of the cells, we examined the distribution of the cytoplasmic protein beta-tubulin, the soluble nuclear protein Lamin B, and the chromatin bound protein Histone 3 (H3). Vpr was abundantly present in three of the four fractions (cytoplasmic, membrane, and soluble nuclear) at relatively equivalent amounts, while Vpr was not abundantly detected within the chromatin fraction. Notably, vorinostat exposure had no detectable effect on the abundance of Vpr in any sub-cellular compartment, indicating that vorinostat does not enhance the transcription-promoting activity of Vpr by modulating Vpr localization (S3 Fig).

### Vpr induces G2/M arrest and apoptosis in primary CD4$^+$ T-cell model

Although the function of Vpr in HIV-1 infection and replication in macrophages is relatively well-characterized [35–38], the role of Vpr in CD4$^+$ T cells is less well understood. Studies have suggested that Vpr induces G2/M arrest and apoptosis in CD4$^+$ T cells [39–41], and some evidence suggests that G2/M arrest may be correlated with increased viral gene expression [28,30]. In our experiments, we observed consistently fewer GFP$^+$ cells in samples infected with HIV-GFP-Vpr versus HIV-GFP, which could potentially have been caused by cell cycle arrest, apoptosis or both. We therefore sought to confirm that Vpr expression induces G2/M arrest and apoptosis in our primary CD4$^+$ T-cell model. Using DAPI, a DNA fluorophore that stoichiometrically binds dsDNA, thereby allowing resolution of cell cycle phases based on the DNA content of each cell, we observed that cells productively infected with HIV-GFP- Vpr had a significantly higher proportion of cells (~46%)

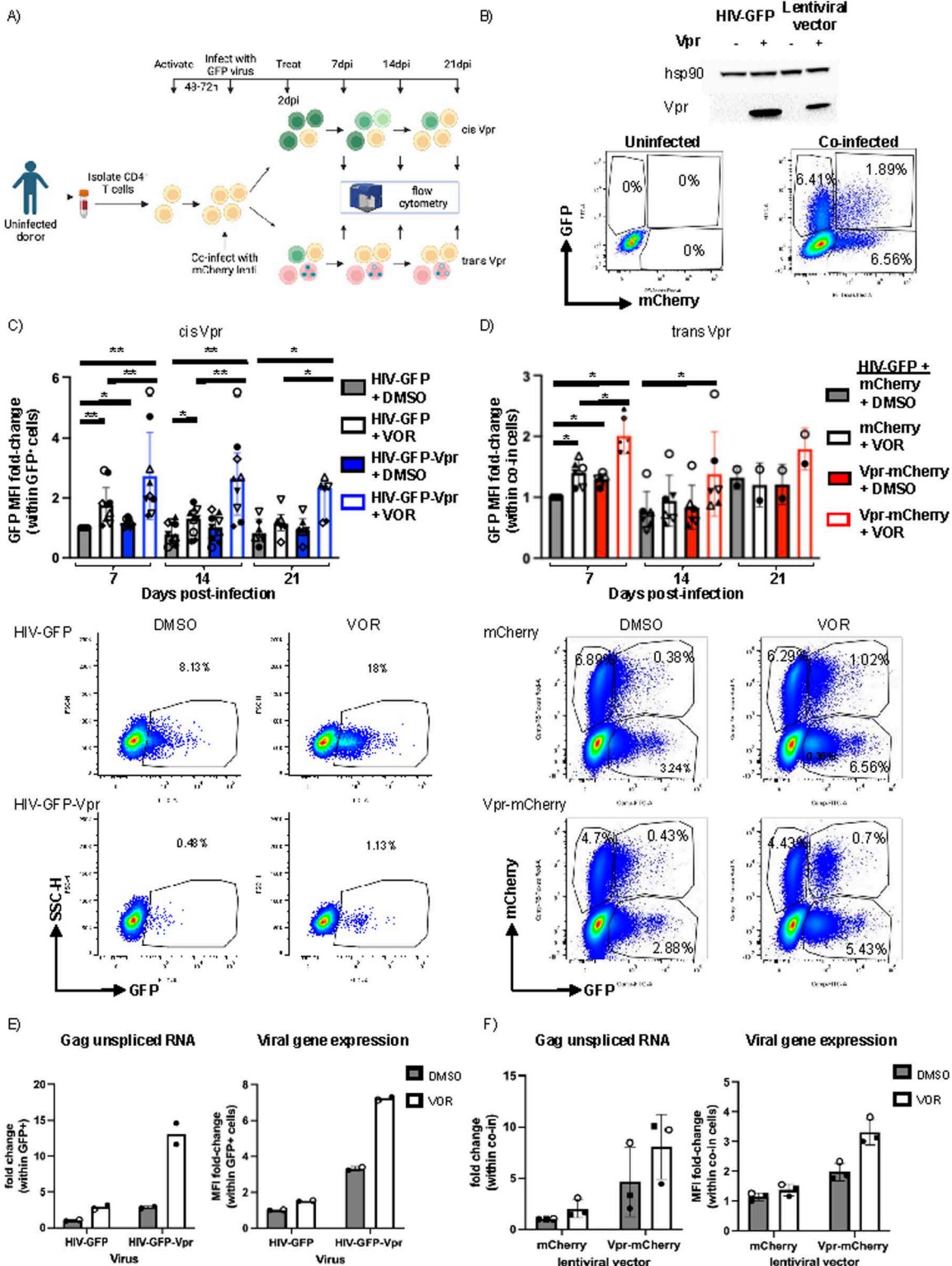

**Fig 2. Viral gene expression within productively infected (GFP⁺) cells is increased in the presence of Vpr and vorinostat (VOR). A)** Top panel (cis Vpr expression): schematic of workflow with infection with Vpr revertant vs. Vpr-null virus (HIV-GFP) and lower panel (trans Vpr expression): co-infection with Vpr-mCherry lentivirus or mCherry lentivirus and HIV-GFP (created with Biorender.com). **B)** Top panel: western blot for Vpr expression

in 293T cells transfected with HIV-GFP, Vpr revertant plasmid, mCherry lentivirus plasmid, or Vpr-mCherry lentivirus plasmid. Lower panel: representative dot plot of co-infected CD4+ T cells. C-D) Representative quantification of GFP MFI over 21d in C) cis Vpr infection model (n = 9) and D) trans Vpr infection model (n = 6). Representative dot plots with and without VOR below. %GFP+ cells is indicated on graph. GFP MFI can be visualized as lateral movement of cells within GFP+ gate. E-F) Left: Representative GFP MFI quantification by flow cytometry and (right) gag unspliced transcript quantification by qPCR in E) cis Vpr infection model (n = 2) and F) trans Vpr infection model (n = 3 donors).

in G2/M phase than cells productively infected with HIV-GFP (~17%) (S4A Fig). This finding suggests that Vpr arrests cell cycle at G2/M in our primary cell infection model. This effect was not observed in uninfected cells. As previously observed, vorinostat also induced G2/M arrest [42–44] in both HIV-GFP-Vpr- and HIV-GFP-infected cells.

To assess apoptosis, we stained infected cells at two days post-infection with Zombie Violet (ZV), a fluorescent amine-binding viability dye that is excluded from live cells, and Annexin V, which binds phosphatidylserine, a phospholipid that is exposed on the surface of apoptotic cells [45]. Notably, we observed elevated levels of both early Annexin V+, ZV-) and late apoptotic/dead cells (Annexin V+, ZV+) in samples infected with Vpr-expressing HIV than those infected with HIV-GFP. This result demonstrates that Vpr induces apoptosis and likely other types of cell death in our primary CD4+ T-cell model as in other HIV-1 infection models [39,46–48] (S4B Fig).

## Combined effect of Vpr and vorinostat on HIV expression depends on the p300-binding domain of Vpr

In addition to inducing G2/M arrest and apoptosis, Vpr has been shown to interact with numerous cellular factors, notably, DDB1-Cul4-associated factor 1 (DCAF1), which targets host factors for proteasomal degradation. Another cellular factor identified by biochemical assay to interact with Vpr is the histone acetyltransferase (HAT) p300 [27,49], a transcriptional co-activator that modulates transcription via chromatin remodeling and by binding to transcription factors, such as NF-κB. To assess the mechanism by which Vpr activates viral gene expression in the presence of vorinostat, we generated a set of four previously characterized Vpr mutants in both the HIV-GFP and mCherry-expressing lentivirus background (S5A Fig). These Vpr mutants were 1) G2/M arrest induction (Y50A) [50], 2) G2/M arrest and apoptosis induction (R90K) [37], 3) DCAF1 binding (Q65R) [51], and 4) p300 binding (F72A/R73A) [27].

We first confirmed Vpr expression for each of the mutant Vpr-expressing viruses by western blot of transfected 293T cells (S5B, S5C Fig). Notably, three of the four mutants (F72A/R73A, Y50A, and R90K) were expressed at a lower level when expressed in cis (encoded by HIV-GFP) but not when encoded within the recombinant Vpr expression lentivirus. To assess whether any of the four Vpr mutants exhibited altered activity in our primary CD4+ T-cell latency model, we then infected activated CD4+ T cells with HIV-GFP, HIV-GFP-Vpr or one of the four HIV-GFP-Vpr mutant viruses and then treated the infected cells with vorinostat or vehicle control (DMSO). When we measured the effect on viral gene expression at 14dpi, we observed that the F72A/R73A (p300-binding) mutant exhibited reduced ability to activate HIV gene expression in the presence of vorinostat (Fig 3A, 3B). By contrast, the Q65R, Y50A and R90K mutants were all still able to induce HIV expression in combination with vorinostat to a similar degree as wild type Vpr (Fig 3A). Interestingly, the F72A/R73A (p300-binding) mutant also had a slightly altered rate of entry into latency (i.e., % GFP+ cells within Thy1.2+ population over time) compared with HIV-GFP-Vpr and the other HIV-GFP-Vpr mutants (S5D Fig). These data suggest that the p300-binding domain of Vpr is required for HDACi-dependent transactivation of viral gene expression. Although expression of the F72A/R73A mutant was reduced in transfected 293T cells relative to wild type Vpr (S5A Fig), the other Vpr mutants with reduced expression (Y50A and R90K) retained full activity for promoting HIV expression when combined with vorinostat, suggesting that this phenomenon is not exquisitely sensitive to Vpr expression levels.

When we examined the activity of these Vpr mutants following co-infection with HIV-GFP and lentiviral-driven Vpr expression in trans (Fig 2A), we also observed a reduced ability of the F72A/R73A mutant to drive viral gene expression in combination with vorinostat (Fig 3B, 3D). Of note, Vpr (F72A/R73A) was expressed at a similar level to wild-type Vpr in lentivirus-transfected 293T cells (S5C Fig), indicating that the reduced ability of this mutant to promote HIV gene

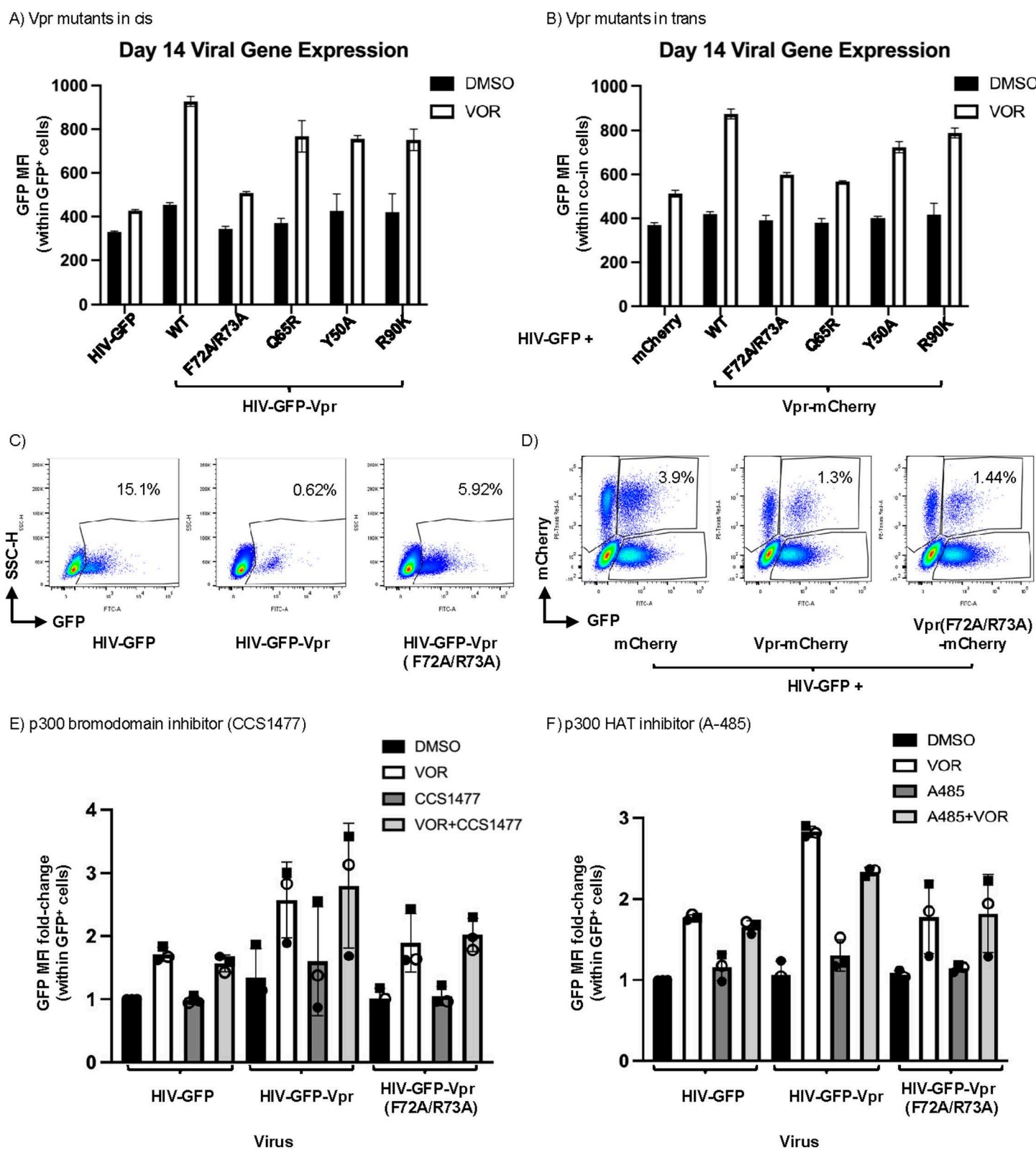

**Fig 3. Vpr p300-binding site and p300 acetyltransferrase activity required for combined effect of Vpr and vorinostat (VOR).** A-B) Day 14 post-infection viral gene expression (GFP median fluorescence intensity; MFI) within DMSO- or VOR-treated CD4$^+$ T cells productively infected (GFP$^+$) with **A)** HIV-GFP or HIV-GFP-Vpr mutant virus or B) co-infected with HIV-GFP and mCherry or Vpr-mCherry mutant lentivirus. Representative of two

donors with three technical replicates each. **C)** Representative dot blots from A) for cells infected with HIV-GFP, HIV-GFP-Vpr, or HIV-GFP-Vpr (F72A/R73A) and treated with VOR. **D)** Representative dot blot from B) for cells co-infected with HIV-GFP and Vpr (F72A/R73A)-mCherry, Vpr-mCherry, or mCherry and treated with VOR. E-F) CD4$^+$ T cells infected with HIV-GFP, HIV-GFP-Vpr, or HIV-GFP-Vpr (F72A/R73A) virus and treated with DMSO, VOR, or E) p300 bromodomain inhibitor (CCS1477) alone or in combination with VOR or F) p300 histone acetyltransferase inhibitor (A485) alone or in combination with VOR. Representative of three donors with three technical replicates each. VOR = vorinostat.

expression was not due to lower Vpr expression. Interestingly, HIV gene expression was also lower in infected cells that were transduced with a lentivirus expressing the Q65R mutant compared with wild-type Vpr (Fig 3B), although this may have been related to the overexpression of Vpr when cells were infected with Vpr (Q65R)-mCherry. Overall, these data indicate that the p300-binding domain of Vpr is required for its ability to promote high HIV-1 expression in the presence of vorinostat.

## A chemical inhibitor targeting the p300 histone acetyltransferase domain inhibits the combined effect of Vpr and vorinostat on HIV-1 expression

Given our observation that the p300-binding domain of Vpr is required for the combined pro-transcriptional effect of Vpr and vorinostat, we next sought to further probe the mechanism of this effect using p300-targeted inhibitors. p300 has five protein-binding domains in addition to a histone or protein acetyltransferase (HAT/PAT) domain and a bromodomain (BRD) that mediates binding to acetylated histones. We first assessed the effect of the p300-specific bromodomain inhibitor CCS1477 [52] on HIV-1 gene expression when Vpr expression is combined with vorinostat treatment. We infected primary CD4$^+$ T cells with HIV-GFP, HIV-GFP-Vpr, or HIV-GFP-Vpr (F72A/R73A). Two days post-infection, we exposed the cells to DMSO, vorinostat, p300 bromodomain inhibitor (CCS1477), or a combination of vorinostat and CCS1477 and monitored GFP expression for 14 days. Notably, we did not observe any effect of CCS1477 alone or in combination with vorinostat on the level of viral gene expression in infected cells (measured by GFP MFI; Fig 3E). Next, to assess the role of the histone acetyltransferase domain of p300 in the combined effect of Vpr and vorinostat, we again performed experiments in CD4$^+$ T-cells infected with HIV-GFP, HIV-GFP-Vpr, or HIV-GFP-Vpr F72A/R73 but instead exposed the infected cells to a p300 HAT inhibitor (A-485). To select an A-485 concentration, we performed a 6-point, 3-fold dose-response curve from 5-0.02µM and selected 560nM as the highest concentration at which cells were still viable. A-485 has been reported to have high selectivity for the p300 HAT domain over other HAT family members and BET bromodomain proteins [53]. We observed that the addition of A-485 attenuated the effect of vorinostat and Vpr to a level similar to that seen with the F72A/R73A Vpr mutant that lacks p300-binding activity (Fig 3F). This result suggests that HAT activity of p300 contributes to the combined strong pro-transcriptional effect of vorinostat and Vpr and that the p300 bromodomain is not required for this interaction.

## Combined vorinostat exposure and Vpr expression induces a unique transcriptional signature in HIV infected cells

Having observed that the combination of vorinostat exposure and Vpr expression enhanced HIV-1 expression beyond each individual condition alone, we next hypothesized that combined vorinostat exposure and Vpr expression might also induce a unique effect on the transcriptome of infected cells. To investigate this hypothesis, CD4$^+$ T cells from three different seronegative donors were infected with either HIV-GFP or HIV-GFP-Vpr and treated with either DMSO or vorinostat for 12 days. As expected, we observed that the combined effect of vorinostat and Vpr expression potently upregulated HIV expression (Fig 4A). We then isolated RNA from sorted infected (GFP$^+$) cells and performed bulk RNA sequencing. When we examined the overall structure of the RNAseq data with principal component analysis (PCA; Fig 4B), we observed that the datapoints could be separated based on condition, indicating a consistent biological signal across donors. We then determined sets of differentially expressed genes (DEGs) using DESeq2 [54]. Using an adjusted p-value of 0.05 as the

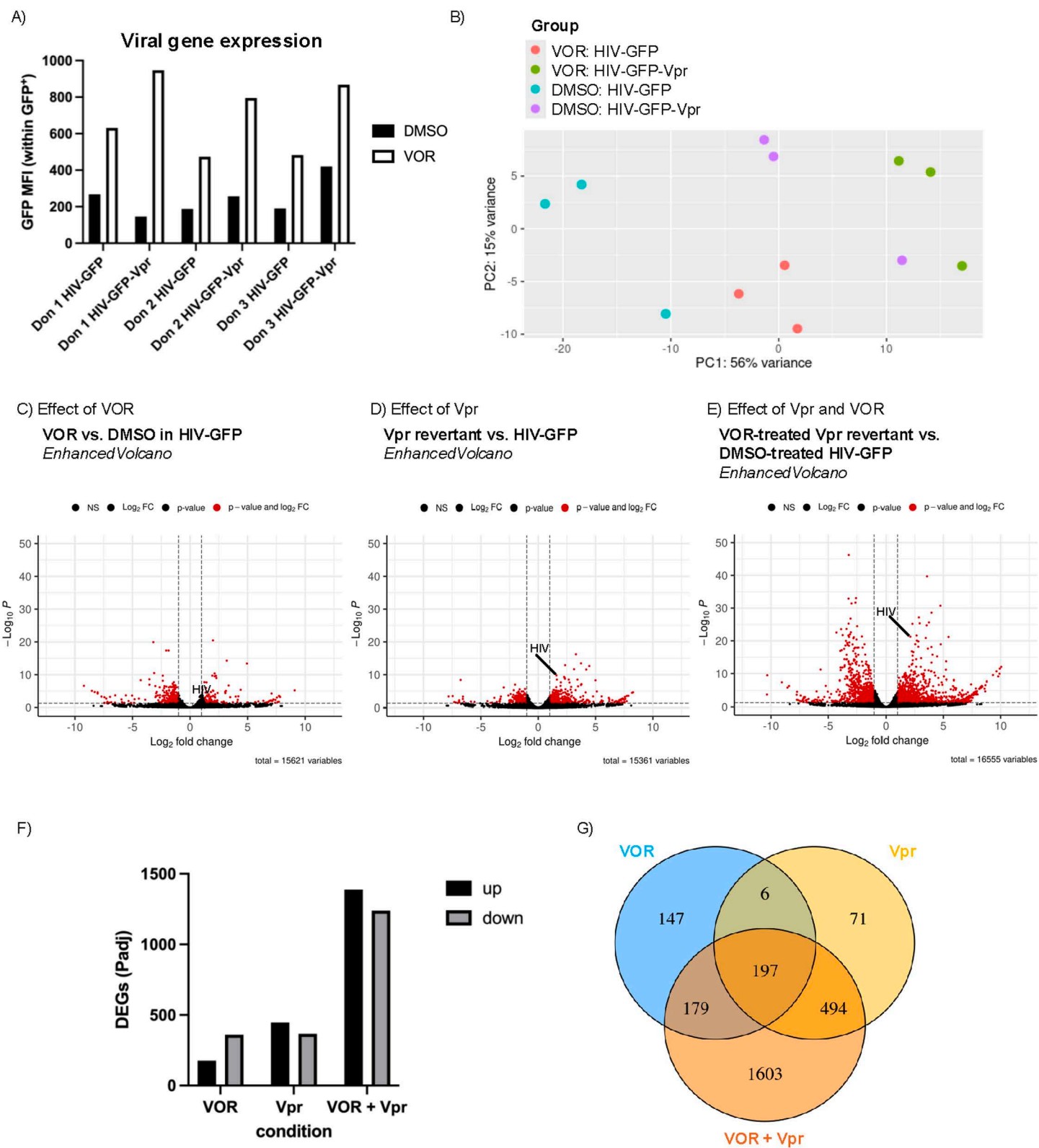

**Fig 4. Many more differentially expressed genes (DEGs) result from VOR treatment and Vpr expression than either condition alone. A)** Primary CD4$^+$ T cells from 3 seronegative donors were infected with either HIV-GFP or HIV-GFP-Vpr and treated with 500nM vorinostat starting 2 days

post-infection (dpi). At 14dpi, cells were analyzed by flow cytometry, and RNA was isolated for bulk RNA sequencing (RNASeq). **A)** Flow cytometry results for GFP median fluorescence intensity (MFI). **B)** Principal component analysis of RNASeq samples from three donors. C-E) Volcano plots of DESeq2 results for **C)** VOR- vs. DMSO-treated cells infected with HIV-GFP, **D)** DMSO-treated cells infected with HIV-GFP-Vpr or HIV-GFP, or **E)** VOR-treated, HIV-GFP-Vpr-infected cells vs. DMSO-treated, HIV-GFP-infected cells. Y-axis is based on adjusted P-value. Alpha = 0.05. **F)** Quantification of DEGs from C-E. **G)** Venn diagram of DEGs from C-E.

significance cutoff, we observed 177 upregulated and 361 downregulated genes in vorinostat- vs. vehicle control-treated cells infected with HIV-GFP. When comparing vehicle control-treated cells infected with HIV-GFP-Vpr vs. HIV-GFP, there were 447 upregulated and 367 downregulated genes. Notably, cells treated with vorinostat and infected with HIV-GFP-Vpr exhibited a total of 1388 upregulated vs. 1239 downregulated genes compared with DMSO-exposed cells infected with HIV-GFP (Fig 4C–4G). These data indicate that the combined vorinostat/Vpr condition induces a much more potent transcriptional signature in HIV-1-infected CD4+ T cells than do vorinostat or Vpr alone. Thus, the potent effect of Vpr/vorinostat is also observed with numerous cellular genes and is not specific to HIV. These observations provide further evidence that Vpr has a strong transcriptional effect in CD4+ T cells that is blocked by HDACs and revealed by vorinostat treatment.

We next examined whether any specific biological pathways were affected by Vpr and vorinostat either alone or in combination. Gene set enrichment analysis (GSEA) using the Hallmark gene set [55] confirmed downregulation of pathways involved in G2/M checkpoint, mitosis, and MTORC1 signaling in the presence of vorinostat, as expected and as we have previously reported [21]. We observed a similar effect for Vpr alone (S6E Fig). Interestingly, IL6/Jak/Stat3 signaling was downregulated in vorinostat- vs. vehicle control-treated HIV-GFP-Vpr-infected cells. In addition, we observed an increase in the expression of genes involved in "TNFα signaling via NF-κB" in vorinostat-treated, HIV-GFP-Vpr-infected cells vs. vorinostat-treated, HIV-GFP-infected cells treated with vorinostat. We did not, however, observe enrichment for any unique pathways in HIV-GFP-Vpr-infected cells treated with vorinostat vs. HIV-GFP-infected cells treated with DMSO. Overall, these data suggest that, while signaling through NF-κB may play a role in the increased HIV-1 transcription in the presence of vorinostat and Vpr, the combined effect of Vpr and vorinostat on host cells is likely broad and involves multiple pathways.

### Vpr expression in HIV-1-infected cells increases the proportion of cells with a central memory phenotype

Given the potent effect of Vpr on the transcriptome of infected cells, we hypothesized that Vpr might affect the T-cell memory subset phenotype of the infected cells. CD4+ T cells in vivo can be subdivided into functionally distinct subsets and differentiate along a linear trajectory from naïve ($T_N$) and stem cell memory ($T_{SCM}$) cells to central memory cells ($T_{CM}$) to effector memory cells ($T_{EM}$) and finally to terminally differentiated effector memory cells ($T_{EMRA}$). We therefore sought to determine whether Vpr modulates memory-cell composition in activated primary CD4+ T cells during latency establishment. Based on the two cell surface markers CCR7 and CD45RA, we defined four populations of memory CD4+ T cells as follows: CD45RA+CCR7+ stem cell memory ($T_{SCM}$), CD45RA-CCR7+ central memory T cells ($T_{CM}$), CD45RA-CCR7- effector memory T cells ($T_{EM}$), and CD45RA+CCR7- effector memory re-expressing RA T cells ($T_{EMRA}$). Because we had previously observed that all cells in experiments using this model are CD95+ [21], we defined CD45RA+CCR7+ cells as $T_{SCM}$ cells rather than $T_N$ cells. Interestingly, at weeks one, two, and three post-infection, we observed a significantly increased fraction of cells with a $T_{CM}$-like phenotype (q < 0.0375 by Wilcoxon matched-pairs, signed-rank test) following infection with HIV-GFP- Vpr compared with HIV-GFP infection (Fig 5A). We also observed an increase in the percentage of $T_{CM}$ cells when the CD4+ T cells were transduced with Vpr-expressing lentivirus but not a control lentivirus (Fig 5B), although the effect was not statistically significant. We did not observe this effect in uninfected cells (S6A, S6B Fig). These data demonstrate that Vpr expression in HIV-1-infected CD4+ T cells alters the composition of the overall population by increasing the percentage of cells with a $T_{CM}$ phenotype.

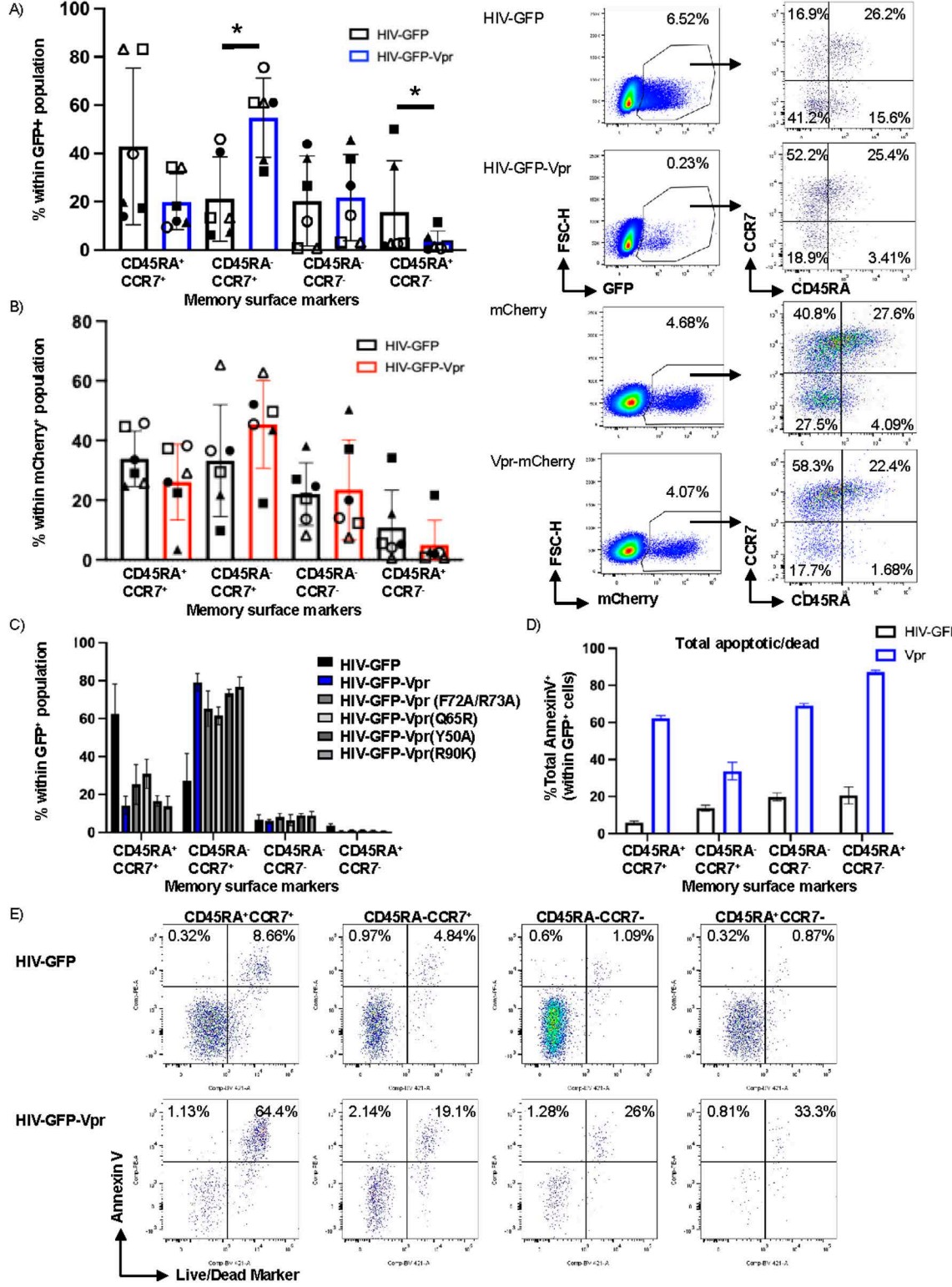

**Fig 5. Vpr expression results in an increased fraction cells with T$_{CM}$-like phenotype (CD45RA$^-$CCR7$^+$).** A-B) Representative quantification of percentage of cells in each of four memory compartments as defined by CD45RA and CCR7 surface marker expression in CD4$^+$ T cells infected with **A)**

HIV-GFP or HIV-GFP-Vpr (n = 6) or B) mCherry or Vpr-mCherry lentivirus (n = 6) with representative dot plots **C)** % cells in each of the four memory compartments following infection with HIV-GFP-Vpr mutant viruses (n = 2). **D)** %Annexin V$^+$ cells within GFP$^+$ CD4$^+$ T cells in each of four CD4$^+$ T-cell memory compartments and E) representative flow plot from each quadrant of CCR7 vs. CD45RA dot plots (n = 5).

We next examined whether the increase in $T_{CM}$ cells within the culture was affected by any of the Vpr mutations that we had previously examined. Interestingly, none of the four Vpr mutations (F72A/R73A, Q65R, Y50A, and R90K) significantly abrogated the ability of Vpr to increase the proportion of $T_{CM}$ cells in the culture (Fig 5C). These data indicate that the effect of Vpr on the memory phenotype of CD4$^+$ T cells is independent of the domains affected by these mutations and is mediated by a distinct mechanism.

### $T_{CM}$ cells are resistant to Vpr-induced apoptosis in HIV-1-infected cells

We hypothesized that the increased fraction of CD4$^+$ T cells with a $T_{CM}$ phenotype following Vpr expression could potentially result from one of two mechanisms: 1) Vpr-induced epigenetic and/or transcriptional reprogramming of CD4$^+$ T cells to a $T_{CM}$ phenotype or 2) Vpr-mediated killing of non-$T_{CM}$ subsets. To distinguish between these possibilities, we measured the level of apoptosis within each T-cell subset ($T_{SCM}$, $T_{CM}$, $T_{EM}$ and $T_{EFF}$) after infection with HIV-GFP-Vpr. To identify cells in different stages of apoptosis we stained infected cells at 14 days post-infection with Annexin V and ZV. Cells in early apoptosis were identified as Annexin V$^+$/ZV$^-$, while cells in late apoptosis/dead were identified as Annexin V$^+$/ZV$^+$. At 14 days post infection, we observed, as expected, an overall increase in cells that were in both late and early apoptosis for cells infected with HIV-GFP-Vpr but not with HIV-GFP. Interestingly, when we divided the data based on T-cell phenotype, we observed different levels of Annexin V$^+$ (apoptotic/dead) cells across T-cell subtypes. In particular, cells with a $T_{CM}$ phenotype had lower levels of total and late apoptosis/dead cells than other subsets (Fig 5D). Interestingly, the level of early apoptotic cells was higher in $T_{CM}$ cells, suggesting that a non-apoptotic pathway may have been responsible for the increased cell death in other subsets or that most of the apoptosis induced in other subsets had already occurred by the time we measured cell death/apoptosis within each subset at days seven and 14. Importantly, the effects were only observed in productively infected (GFP$^+$) cells and were not observed in GFP$^-$ cells within the same culture (S6A–C Fig), demonstrating a cell-intrinsic mechanism for this killing. These data suggest that the ability of Vpr to induce apoptosis/cell death is attenuated in $T_{CM}$ cells compared with other major CD4$^+$ T-cell subsets.

To further investigate the effect of Vpr on apoptosis-regulating pathways in infected cells, we also re-examined our bulk RNASeq data from CD4$^+$ T cells infected with a Vpr-expressing HIV strain for relative expression of a previously defined array of 44 cell death genes involved in apoptosis, necroptosis, and pyroptosis [56]. We hypothesized that, by inducing apoptosis, Vpr expression could potentially select for cells with different levels of expression of these cell death/survival-regulating genes. When we compared the transcriptional profiles of cells infected with HIV-GFP-Vpr versus cells infected with HIV-GFP, we observed eight cell death-regulating genes that were differentially expressed: BCL2L11, BIK, BAX, BIRC2, BIRC3, ZBP1, GSDME, and PARP1. We observed upregulation of the necroptosis gene ZBP1, the pro-apoptosis gene BCL2L11, and the pro-pyroptosis gene GSDME in Vpr expressing cells, and we observed downregulation of the pro-apoptotic genes BIK and BAX and upregulation of the caspase antagonists BIRC2 and BIRC3 (Table 1) in Vpr expressing cells. We speculate that increased expression of BIRC3 and BIRC3 results from selection for cells that express higher levels of these genes. Overall, these data support the hypothesis that the observed enrichment in $T_{CM}$ cells after infection with a Vpr-expressing virus may be due to their ability to resist Vpr-induced apoptosis. Thus, Vpr expression during latency establishment may help to enrich the initial pool of infected cells that enter the reservoir within the $T_{CM}$ compartment.

## Discussion

Recently, it has been appreciated that a sizable fraction of the long-lived latent reservoir is established or stabilized around the time of ART initiation [19,20,57]. Understanding how host and viral factors regulate HIV-1 expression as a

**Table 1. DEGs in HIV-GFP-Vpr- vs. HIV-GFP-infected cells within a previously published subset of cell death pathway-related genes [56].**

| DEGs in HIV-GFP-Vpr- vs. HIV-GFP-infected cells within a previously published subset of cell death pathway-related genes [56]. Pathway | SubPathway | Gene Name | Protein Name | HIV-GFP-Vpr vs. HIV-GFP | padj | not adj |
|---|---|---|---|---|---|---|
| APOPTOSIS | PROAPOPTOTIC BH3-ONLY | BCL2L11 | BIM | 1.157866 | 0.049247 | 0.002582 |
| APOPTOSIS | PROAPOPTOTIC BH3-ONLY | BIK | BIK | -1.215119 | 0.249194 | 0.034951 |
| APOPTOSIS | PROAPOPTOTIC EFFECTOR | BAX | BAX | -0.5093965 | 0.221866 | 0.028815 |
| APOPTOSIS | CASPASE ANTAGONIST | BIRC2 | cIAP1 | 0.8498609 | 0.054505 | 0.002991 |
| APOPTOSIS | CASPASE ANTAGONIST | BIRC3 | cIAP2 | 1.582484 | 0.000994 | 1.04E-05 |
| NECROPTOSIS | NECROPTOSIS | ZBP1 | ZBP1 | 1.142353 | 0.012517 | 0.000362 |
| PYROPTOSIS | PYROPTOSIS | GSDME | GSDME | 1.3613 | 0.285323 | 0.044727 |
| PARTHANATOS | PARTHANATOS | PARP1 | PARP1 | -0.453507 | 0.242379 | 0.033514 |

Padj = adjusted p-value; not adj = p-value.

provirus enters a transcriptionally silent state as the infected cell enters the latent reservoir will lead to identification of candidate therapeutic targets for a latency prevention approach. Whereas factors that maintain proviral quiescence and latency have been studied extensively, the mechanisms involved in latency establishment are less well understood and may differ in some respects from those that maintain latency. We hypothesize that these differences exist both because the host epigenetic and transcriptomic landscapes differ between the two states and because, unlike latent provirus, proviruses undergoing transcriptional downregulation still initially produce viral proteins that can interact with host factors.

We previously identified HDACs as critical early factors for viral entry into latency, both in terms of transcriptional silencing and for differentiation into T-cell subsets typically enriched for latent virus [21]. This prior work focused on host cell factors that regulate HIV-1 silencing in CD4+ T cells with the goal of identifying potential cellular targets for interfering with or preventing latency from being established. However, because these experiments relied primarily on a reporter strain of HIV-1 with mutations in the regions encoding the HIV-1 structural and accessory proteins, the role of most HIV-1 genes in transcriptional silencing of HIV-1 in CD4+ T cells remained unclear. If specific viral proteins that play important roles in latency establishment can be identified, these proteins could also be targeted along with host cell pathways to disrupt latency.

In this study, we identify Vpr as a key viral protein that plays a role in regulating the establishment of a pool of latently infected CD4+ T cells and that shapes key characteristics of these cells. Using single gene revertant strains of HIV-1 for Gag/Pol, Vpr, Vpu, Vif, and Nef in a primary CD4+ T-cell model of HIV-1 latency establishment, we showed that, while Vpr expression alone does not have a strong effect on viral transcription, Vpr expression in combination with exposure to the class I HDACi vorinostat leads to a dramatic, sustained increase in viral gene expression. We showed that an HDAC1/2i but not an HDAC3i had a similar effect (S2A Fig), suggesting that HDAC1 and/or 2 play a role in blocking the pro-transcriptional activity of Vpr. Interestingly, the combined effect of vorinostat and Vpr was not observed when vorinostat was delivered to latently infected cells (S2B, S2C Fig), likely because Vpr was no longer expressed at sufficient levels in these cells or because HDAC activity is low in resting CD4+ T cells.

Using Vpr mutants and p300 inhibitors, we showed that the combined effect of Vpr and vorinostat depended on a p300-binding domain of Vpr as well as the HAT activity of p300. Additionally, we found that combined Vpr expression and vorinostat exposure induces a potent and unique transcriptional signature in infected cells, further demonstrating the interaction of these pathways. Overall, our results suggest that, as HIV-1 enters latency, Vpr has strong pro-transcriptional

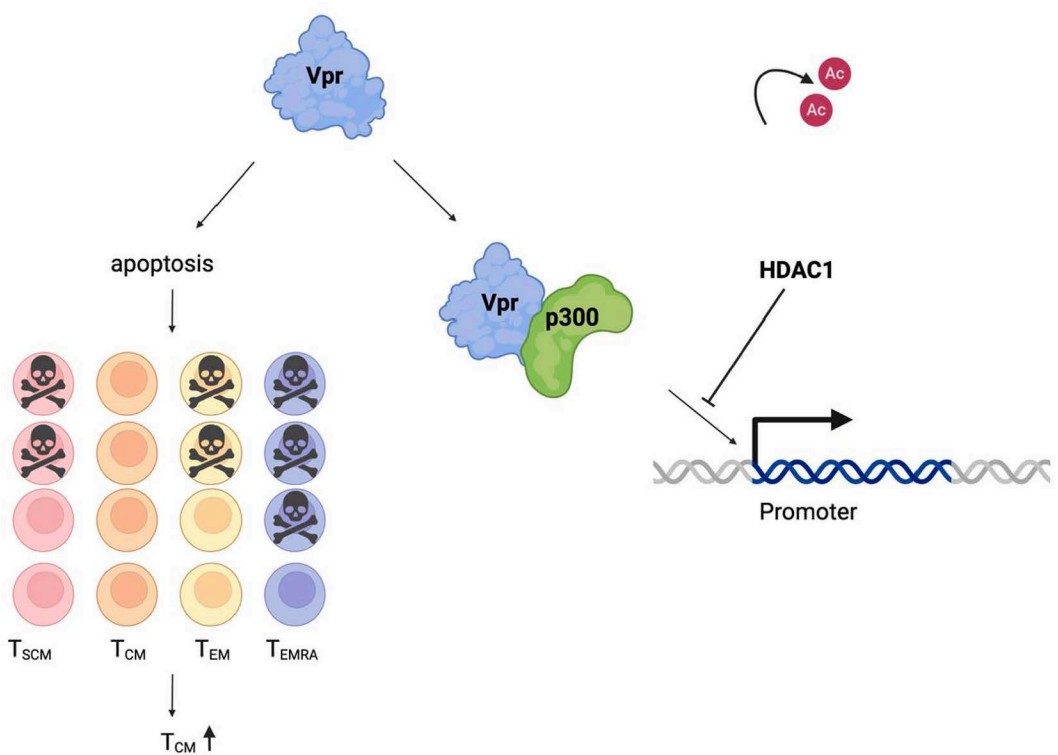

**Fig 6. Model of role of Vpr in HIV-1 transcription and T-cell memory subset composition.** Left: Vpr in memory CD4+ T-cell subset composition. Vpr expression leads to enrichment for $T_{CM}$-like cells (CD45RA-CCR7+), likely because of selective cell death in other memory compartments. Right: Vpr in HIV-1 transcription. Vpr expression leads to transactivation of the HIV-1 promoter (long terminal repeat; LTR), either directly or indirectly, in a manner dependent on an intact p300-binding domain as well as an intact p300 acetyltransferase domain. Histone acetyltransferase activity blocks this pro-transcriptional activity of Vpr. Created with Biorender.com.

activity that requires HAT activity of p300 but is blocked by HDACs (Fig 6). Establishment of HIV latency thus results, in significant part, from an HDAC-mediated restriction to Vpr activity in CD4+ T cells.

Previous studies have indicated a pro-transcriptional activity for Vpr [27,28,30,49,58,59]. However, the mechanism involved is unclear and has been attributed to multiple functions of Vpr, most commonly cell cycle arrest [28,30]. Studies have also reported a link between Vpr-induced cell cycle arrest and DCAF1-binding and pro-apoptotic activity, making it difficult to distinguish each function's individual contribution to viral promoter transactivation. Vpr has also been shown to transactivate HIV-1 gene expression via p300, both directly and indirectly [27,49,60], although this function is less well-studied. Many of these previous Vpr studies were performed in immortalized cell lines. Although cell lines have been an invaluable tool for studying many aspects of HIV-1 virology and pathogenesis, they have been modified to evade cellular senescence, making them unlikely to be the most accurate model of an HIV-1-infected cell returning to rest. Further, previous studies have often relied solely on lentiviral expression vectors with strong promoters driving Vpr and/or expression during HIV-latency. This can lead to nonbiological levels of Vpr expression in an epigenetic or transcriptional landscape in which viral protein would not be produced in sizable amounts. To address these challenges, we infected primary CD4+ T cells with virus in which Vpr expression was driven by the viral promoter or observed the action of exogenous Vpr on the viral promoter.

The p300 HAT domain has previously been shown to activate HIV-1 gene expression via several possible mechanisms, including histone acetylation-mediated chromatin remodeling and acetylation of the viral protein Tat and host transcription

factors such as NF-κB [61]. Tat acetylation is required for Tat-mediated transactivation of the HIV-1 promoter. Acetylation of NF-κB subunits enables NF-κB to bind with more stability to NF-κB-binding sites and to the basal transcription machinery, promoting HIV-1 gene expression [61,62]. However, further study, particularly of chromatin changes at the viral promoter as well as promoter occupancy, will be required to better understand the role of Vpr, HDACs, and p300 HAT activity in regulating HIV-1 gene expression, particularly as the virus enters transcriptional latency. These studies will be challenging because the toxicity of Vpr limits the number of cells available for analyses. Ultimately, understanding how Vpr regulates HIV-1 transcription in the context of host epigenetic and transcription factors will aid in the development of agents that can prevent latency formation or more potently reverse latency. Identifying the individual HDAC that counteracts Vpr's pro-transcriptional activity in T cells will also help inform therapeutic approaches to targeting the reservoir.

Following cis or trans Vpr expression, we also observed an increase in cells with a $T_{CM}$-like phenotype (CCR7$^+$CD45RA$^-$; Fig 5A, 5B), a long-lived cell type in which a significant fraction of the latent reservoir is thought to reside [23,63,64]. Of note, the effect of cis Vpr was stronger than the effect with trans Vpr. Because Vpr is packaged within the HIV-1 virion, our "cis" model also has some initial trans delivery of Vpr as well as LTR-driven Vpr expression, offering one explanation for why the effect with cis Vpr may be stronger. Overall, our observation that $T_{CM}$-like cells are increased in the presence of Vpr suggest that Vpr, which is present from the early stages of infection [65,66], could play an important role in shaping the composition of the latent reservoir. This result also offers an alternative, indirect model for the role of Vpr and vorinostat in HIV-1 gene expression. Our group has observed in our model that LRAs induce GFP expression to a greater extent in $T_{CM}$-like cells than in other cell types, likely because GFP expression is subject to less restriction in these cell types [67]. Therefore, the increase in HIV-1 gene expression in the presence of Vpr and vorinostat may also be because Vpr enriches for $T_{CM}$-like cells, which in turn are more inducible by vorinostat.

Although our data are consistent with a "selection" model for how Vpr increases the relative frequency of $T_{CM}$ cells in the latent pool, it is also possible that Vpr reprogramming of CD4$^+$ T cells into $T_{CM}$-like cells occurs. Previous work has shown that the majority of the early transcriptional [68] as well as the proteomic changes [69] that occur following HIV-1 infection are driven by Vpr. In addition, in resting CD4$^+$ T cells infected by cell-to-cell transmission, Vpr reprogrammed cells to a tissue resident memory phenotype [70]. However, in our bulk RNASeq dataset, none of the pathways enriched by GSEA using the Hallmark gene set would clearly lead to memory cell reprogramming (S7E Fig). Our data instead support a model in which $T_{CM}$ cells are more resistant to Vpr-induced apoptosis/cell death than other memory cell subsets, leading to selective enrichment of infected $T_{CM}$ cells in the overall pool of infected cells (Fig 6). Indeed, previous work has shown that $T_{CM}$ cells are less susceptible to cell death following HIV reactivation [71,72]. Interestingly, we did not observe a change in the enrichment for $T_{CM}$-like cells following infection with any of the four Vpr mutants generated, including the mutant that lacks apoptosis-inducing ability (Fig 5C), indicating that the preferential survival of $T_{CM}$ cells may be a complex phenomenon. We also speculate that, when viral gene expression from latent proviruses is reignited, either sporadically or after LRA-mediated reactivation, infected $T_{CM}$ cells are preferentially protected from Vpr-mediated cell death, allowing these cells to persist despite effective latency reversal. Indeed, even LRAs that have shown potent latency-reversing activity in vivo have had no apparent effect on the size of the reservoir [15,73–75]. Thus, the resistance of $T_{CM}$ cells to Vpr-mediated killing could represent a key mechanism of viral persistence.

Interestingly, we did not observe increased apoptosis/cell death in uninfected cells in our model (S4A, S4B Fig) and, correspondingly, we did not see enrichment for a $T_{CM}$-like phenotype in uninfected cells. The CD4$^+$ T-cell depletion that occurs in HIV-1 infection largely results from bystander death [76,77], an effect that has been attributed to Vpr release from infected cells and uptake by neighboring cells [46]. However, studies examining Vpr-induced cell death in bystander cells have used extracellular Vpr peptide added to cell cultures, rather than examining the effect of co-culturing infected cells with uninfected cells [78–80]. Still, given that our model employed purified CD4$^+$ T cells and a nonreplicative virus, it is possible that when Vpr is exposed to other cell types or when new virion is produced, Vpr induces some apoptosis in neighboring cells.

The mechanism by which $T_{CM}$ cells resist Vpr-induced apoptosis/cell death is still unclear. In our bulk RNASeq data, we observed an upregulation of two caspase antagonists (BIRC2, BIRC3) as well as downregulation of two pro-apoptotic proteins (BIK, BAX) in cells that had survived infection with a Vpr-expressing virus for two weeks, while a different apoptosis gene (BCL2L11), a pro-necroptosis pathway gene (ZBP1), and a pro-pyroptosis gene (GSDME) were upregulated (Table 1). This differential gene expression suggests that Vpr induces necroptosis, pyroptosis, and apoptosis in our model but that $T_{CM}$ cells may express higher levels of caspase antagonists, which protects them from apoptosis and other forms of cell death. Of note, pyroptosis has previously been proposed to be a mechanism responsible for bystander cell death in HIV-1, although Vpr has not been specifically implicated in this effect [77]. In addition, a previous study implicated the BIRC2 and BIRC3 protein products cIAP1 and cIAP3, respectively, in protecting macrophages from Vpr-induced cell death during HIV-1 infection [81].

It is important to note that the observed GSMDE and BIRC2/3 upregulation in our study is from bulk RNA sequencing. Additional study on sorted memory cell populations or single-cell RNA sequencing may therefore be informative. One new approach to targeting persistent, long-lived memory cell populations has been to sensitize cells to apoptosis by using the BCL-2 antagonist venetoclax [72]. However, in cells from PWH treated with venetoclax ex vivo, $T_{CM}$ cells were largely spared, suggesting that $T_{CM}$ cells possess unique mechanisms of resisting apoptosis. Understanding how Vpr affects survival and how the latent reservoir forms has important implications both for potentially preventing the effector-to-memory transition that occurs during ART initiation and for targeting latently infected memory cells for cell death [71,82].

## Limitations of the study

We acknowledge several limitations of the study. First, the use of single gene revertant viruses enables the effects of individual proteins on gene expression and CD4$^+$ T-cell memory subset composition to be separated but did not allow us to study the combined effect of HIV-1 Gag/Pol and accessory proteins. We also observed significant overlap in the phenotypes of our mutant Vpr viruses, for example, cell death/apoptosis was not only decreased in the R90K mutant but also to some extent in the other mutants as well. It is possible that future work would benefit from including additional Vpr mutants. However, studies have shown that the functions of Vpr may be interrelated [39,51,83], suggesting that it may be challenging to generate single-function Vpr mutant viruses. It is also possible that p300 inhibitors used were not specific to Vpr-p300 interaction and some of the observed effect on transcription may have been Vpr-independent. However, the lack of effect on GFP expression in either HIV-GFP- or HIV-GFP-Vpr (F72A/R73A)-infected cells indicates though that the abrogation of HIV expression in HIV-GFP-Vpr-infected cells following treatment with p300 HAT inhibitor is Vpr-dependent. Finally, for all experiments, cells were activated ex vivo with anti-CD3/CD28 antibodies and maintained in culture under IL-2/IL-7 conditions, which is likely a stronger stimulation than most T cells experience in vivo and could alter memory formation. In addition, only two surface markers (CCR7 and CD45RA) were used to define memory cell subset. Future work will benefit from including additional surface markers and/or performing functional assays.

## Methods and materials

### Plasmids and viruses

We modified a previously published HIV reporter virus NL4–3-D6-dreGFP [24], herein referred to as HIV-GFP. This strain of HIV contains inactivating mutations in regions encoding all viral proteins except Tat and Rev. In the parental plasmid, the *nef* gene is truncated and point mutations have been introduced to generate premature stop codons in *gag*, *vif*, *vpr*, and *vpu*. A portion of the envelope has also been replaced with destabilized eGFP (dreGFP), and we previously inserted an IRES-Thy1.2 cassette immediately downstream of dreGFP [21] (Fig 1B). Because Thy1.2 has a significantly longer half-life than dreGFP, these two markers can be used to differentiate between productively infected cells (Thy1.2$^+$GFP$^+$) and infected cells that have recently downregulated HIV expression (Thy1.2$^+$GFP$^-$). We used site-directed mutagenesis to

individually revert the stop codons in *gag/pol, vif, vpr, vpu.* To restore *nef* expression, we digested the HIV-GFP plasmid with BamH1 and Xho1 and inserted a gBlock (IDT) with the missing *nef* sequence to restore the complete open reading frame. Lentiviral expression vectors (Addgene 21373) were generated by replacing the GFP reporter gene with mCherry and inserting Vpr coding sequence into the multiple cloning site. Mutated Vpr constructs (Y50A, R90K, Q65R, F72A/R73A) in both the HIV-GFP and lentiviral expression vector backgrounds were generated using NEB assembly. Full plasmid sequences of Vpr mutant plasmids were confirmed by nanopore sequencing.

### Virus production

Virus stocks were generated by transfecting HEK293T/17 cells (ATCC) with packaging plasmids psPAX2 and MD2-VSV-G in combination with the relevant viral backbone plasmid using Mirus LT1 transfection reagent at a 3:1 volume to mass ratio. DMEM (Gibco; supplemented with 10% fetal calf serum (FCS) and 1% Penicillin Streptomycin) was replaced with RPMI R10 (10% FCS; 1% Penicillin Streptomycin; 10mM HEPES; 2mM L-Glutamine; 1mM Sodium Pyruvate) 24h post-transfection, and virus-containing supernatant was collected 48h post-transfection. Supernatant was then centrifuged at low speed to remove cellular debris, filtered through a 0.45µm low protein-binding filter, and frozen at -80°C before use.

### Primary CD4+ T-cell infection model and cell culture

CD4+ T cells were obtained from leuko-reduction 769 blood products (STEMCELL Technologies) by gradient centrifugation followed by magnetic isolation and cryopreserved in freezing medium (90% FCS, 10% DMSO) as previously described [21]. Cells were thawed and activated using anti-CD3/anti-CD28 Dynabeads (Gibco) in RPMI R10 supplemented with 100U/mL IL-2 and 1ng/mL IL-7 (PeproTech). After 48h, the activating beads were removed, and the cells were infected by resuspending in media containing virus stocks with 4µg/mL polybrene (Hexadimethrine Bromide) and then spinoculated for 2h at 600g. Viral supernatant was then replaced with RPMI at a concentration of $10^6$ cells/mL. Active infection (%GFP+ cells within Thy1.2+gate) was quantified by flow cytometry 48-72h after infection. Media containing appropriate compound (500nM vorinostat, DMSO vehicle control, or p300 inhibitor as detailed below) was replaced every 2–3 days, and cell density was increased to $2x10^6$ cells/mL after 7 days. Productive infection (%GFP+ within Thy1.2+), viral gene expression (median fluorescence intensity; MFI of productively infected cells), memory phenotype (by CCR7 and CD45RA expression), apoptosis (Annexin V and Zombie-NIR), and cell cycle 4′,6-diamidino-2-phenylindole (DAPI) were measured by flow cytometry every 7 days (Fig 1A). For the latency reversal experiment, cells infected, cultured, and treated with DMSO as above were treated with vorinostat (500nM) for 24h at 14 days post-infection. After 24h, %GFP+ cells and GFP MFI within GFP+ cells were measured by flow cytometry.

### Flow cytometry

Cells were washed with phosphate buffered saline (PBS) and stained with Live-Dead viability dye (Zombie Violet or Zombie-NIR, BioLegend) and an anti-CD90.2 antibody (Thy1.2, BioLegend) in PBS. For experiments assessing cell surface expression of memory markers, cells were then washed with FACS buffer (PBS with 2% 776 FCS, 1mM EDTA) and stained with antibodies to CD45RA and CCR7 (BioLegend). For experiments assessing apoptosis, cells were stained with PE-conjugated Annexin V in Annexin V binding buffer (BioLegend) and analyzed by flow cytometry without fixation. For all other experiments, cells were fixed in 4% paraformaldehyde (PFA) prior to acquisition on a Fortessa flow cytometer (Becton Dickson). Data were analyzed using FlowJo (version 10.1). To quantify cells in G2/M phase, cells were stained post-fixation with DAPI (BioLegend) diluted to 3µM in 1x FOXP3 perm buffer (BioLegend). For GFP quantification, at least 10,000 live cells were collected for analysis. To assess memory markers, at least 50,000 live cells were recorded. Compensation controls were prepared using single-color stained cells. To compensate for the viability stain and CD90.2 antibody, ArC amine reactive compensation beads and UltraComp eBeads Plus compensation beads were used, respectively (Invitrogen). Positive events were gated using fluorescence-minus-one and biological (uninfected cells) controls.

## Gag transcript quantification

Fourteen days post-infection (14dpi) with either HIV-GFP or the HIV-GFP-Vpr revertant, GFP⁺ CD4⁺ T cells from two donors, treated with either DMSO or vorinostat, were sorted into RLT⁺ lysis buffer (Qiagen). RNA was isolated using RNeasy Plus Mini Kit (Qiagen 74134), and Gag transcript was quantified by RT-qPCR on a QuantStudio 3 Real-Time PCR system (Thermo Fisher Scientific) using $\beta$-actin as a reference gene. A commercially purchased Taqman primer/probe set was used for $\beta$-actin (Thermo Fisher Scientific Hs01060665_g1). Double delta CT analysis was used to calculate fold-change in *Gag* transcript levels.

## Selective HDAC and p300 inhibitors

Merck patents and publications provide information on the synthesis and characterization of the selective HDAC1(/2) and HDAC3 inhibitors used in this study [84–86]. We have previously published work with additional information on these inhibitors as well [21]. For experiments with these inhibitors, we treated cells three times a week starting two days post-infection with concentrations near their $EC_{50}$, as previously: 800nM HDAC1/2i and 3µM HDAC3i.

Two different p300 inhibitors were used: bromodomain inhibitor (CCS1477, Selleckchem) and histone acetyltransferase inhibitor (A-485, Selleckchem). Prior to experiments in infected cells, dose curves were performed to assess toxicity and cell viability. Based both on viability and previously reported $EC_{50}$, 0.122µM CCS1477 and 560nM A-485 were selected [52,53].

## Western blotting

To generate whole cell protein lysates, HEK293T/17 cells were harvested and lysed with RIPA buffer (Thermo Fisher) supplemented with 1x protease inhibitors (Roche) and 1% Pierce Universal Nuclease (Thermo Fisher). Protein quantification was performed using DC Protein Assay (Bio-Rad). Proteins were resolved by Tris-Glycine or Tris-Acetate SDS-PAGE depending on protein size and transferred to nitrocellulose membrane. Membranes were blocked in 5% milk in 1x Tris-Buffered Saline (TBS) and probed at 4°C overnight with antibodies (Vpr, Proteintech 51143-1-AP; Hsp90, Cell Signaling Technology 4877) diluted in TBS containing 5% BSA or milk. Membranes were washed with TBS containing 0.1% Tween20 (TBST), probed with secondary antibody (1:10,000) conjugated to horseradish peroxidase for 1h at room temperature, washed again with TBST, and developed using enhanced chemiluminescence (Thermo Fisher).

## RNA sequencing

RNA isolation was performed with RNeasy Plus Mini Kit (Qiagen 74134), and cDNA synthesis and library prep were performed using SMART-Seq mRNA LP (Takara 634768). Libraries were quality controlled and quantified by Qubit dsDNA assay (Thermo Fisher) and capillary electrophoresis (Agilent Tapestation) prior to pooling and 2x50 paired-end sequencing on an Illumina NextSeq 2000 system. Quality reports were generated by fastQC v 0.11.9, and reads were aligned to the GRCh38 human genome as well as a previously generated custom HIV reference genome containing the HIV-GFP sequence using STAR v2.7.10b. Differential gene expression was assessed in R using DESeq2 v1.40.2 [54].

## Statistics

Statistical analyses were performed in GraphPad Prism and R. For all bar graphs, plots indicate mean and standard deviation. Statistical details of experiments can be found in figure legends and results sections. To analyze differences in GFP MFI and within memory subsets between HIV-GFP-Vpr and HIV-GFP, no assumptions of normal data distributions were made, and Wilcoxon matched-pairs signed-rank test was employed for studies that were sufficiently powered (n > 6). Benjamini and Hochberg correction with a false-discovery rate of 0.05 was used. Statistical analysis was not performed for experiments that were not sufficiently powered (n < 6).

## Supporting information

**S1 Fig.  HIV-GFP revertant viruses.** A) Western blot for Vif, Vpr, and p24 of 293T cells transfected with HIV-GFP revertant plasmids. B) CD4 expression in CD4$^+$ T cells infected with HIV-GFP revertant viruses. C) Quantification of GFP$^+$ CD4$^+$ T cells within Thy1.2$^+$ population for each virus. D) Representative flow plots for infection with each revertant virus. GFP$^+$ cells were gated within Thy1.2$^+$ cells.
(TIF)

**S2 Fig.  The role of histone deacetylase inhibitors in latency prevention and reversal.** A) GFP median fluorescence intensity (MFI) within GFP$^+$ cells in HIV-GFP- or HIV-GFP-Vpr-infected cells treated with histone deacetylase inhibitor (HDACi) for 14 days. Results are normalized to GFP MFI within HIV-GFP-infected cells for each condition. B-C) Latency reversal versus latency prevention: Cells infected with HIV-GFP or HIV-GFP-Vpr were cultured for 14 days as in A). After 14 days (14dpi), DMSO-treated cells were treated with VOR (vorinostat) for 24h before B) %GFP$^+$ cells and C) GFP MFI within GFP$^+$ cells was assessed by flow cytometry.
(TIF)

**S3 Fig.  Vpr localization within cell compartments does not change following vorinostat treatment.** Western blot with indicated antibody on cell fractions from cells treated with DMSO or vorinostat. VOR = vorinostat. Sol nuc = soluble nuclear fraction.
(TIF)

**S4 Fig.  Induction of cell cycle arrest and apoptosis/cell death by Vpr.** A) Left: example histogram of DAPI staining for cell cycle. Right: % cells in G2/M in each condition 7 days post-infection. B) Representative flow plots of cells stained with Annexin V and Zombie Violet to measure apoptosis and cell death 2 days post-infection.
(TIF)

**S5 Fig.  Vpr expression in functional mutant viruses.** A) Schematic of Vpr domains adapted from Guenzel et al., 2014 [87]. Induction of cell cycle arrest and apoptosis/cell death by Vpr functional mutant viruses. Vpr expression in 293T cells transfected with mutant B) HIV-GFP-Vpr and C) Vpr-mCherry plasmids.
(TIF)

**S6 Fig.  Memory subsets.** Representative data for %GFP$^-$ (uninfected) cells in each subset as defined by CCR7 and CD45RA in samples infected with A) HIV-GFP-Vpr or HIV-GFP or B) Vpr-mCherry or mCherry lentivirus. C) Representative plot of %Annexin V cells in each subset from cells infected with HIV-GFP-Vpr or HIV-GFP. D) Heat map of differentially expressed genes (DEGs) in GFP$^+$ CD4$^+$ T cells infected with HIV-GFP-Vpr or HIV-GFP (n = 3). E) gene-set enrichment analysis for all DEGs.
(TIF)

## Acknowledgments

We would like to thank Kimberly Enders at the Center for AIDS Research at UNC for her guidance with statistical analysis (P30-AI-050410). We would also like to thank Anne-Marie Turner and Joshua Fox at the Institute for Global Health and Infectious Diseases for performing Oxford Nanopore sequencing on our Vpr mutant plasmids. Finally, we would like to acknowledge the UNC Flow Cytometry Core Facility (RRID:SCR_019170).

## Author contributions

**Conceptualization:** Catherine A. Lewis, Edward P. Browne.

**Data curation:** Catherine A. Lewis.

**Formal analysis:** Catherine A. Lewis.

**Funding acquisition:** David M. Margolis, Edward P. Browne.

**Investigation:** Catherine A. Lewis.

**Methodology:** Catherine A. Lewis, Edward P. Browne.

**Project administration:** Edward P. Browne.

**Resources:** David M. Margolis, Edward P. Browne.

**Software:** Catherine A. Lewis.

**Supervision:** David M. Margolis, Edward P. Browne.

**Validation:** Catherine A. Lewis.

**Visualization:** Catherine A. Lewis.

**Writing – original draft:** Catherine A. Lewis.

**Writing – review & editing:** Catherine A. Lewis, David M. Margolis, Edward P. Browne.

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
