## [Decision Letter · Decision Letter 0]

29 Apr 2025

HIV-1 accessory protein Vpr possesses a cryptic p300-dependent transcription-promoting activity that is blocked by histone deacetylases in CD4+ T cells.

PLOS Pathogens

Dear Dr. Lewis,

Thank you for submitting your manuscript to PLOS Pathogens. After careful consideration, we feel that it has merit but does not fully meet PLOS Pathogens's publication criteria as it currently stands. Therefore, we invite you to submit a revised version of the manuscript that addresses the points raised during the review process.

Please submit your revised manuscript within 60 days Jun 28 2025 11:59PM. If you will need more time than this to complete your revisions, please reply to this message or contact the journal office at plospathogens@plos.org. Please include the following items when submitting your revised manuscript:

We look forward to receiving your revised manuscript.

Kind regards,

David T. Evans

Academic Editor

PLOS Pathogens

Richard Koup

Section Editor

Editor-in-Chief

PLOS Pathogens

orcid.org/0000-0003-2946-9497

Editor-in-Chief

PLOS Pathogens

orcid.org/0000-0002-7699-2064

**Journal Requirements:**

1) We do not publish any copyright or trademark symbols that usually accompany proprietary names, eg ©,  ®, or TM  (e.g. next to drug or reagent names). Therefore please remove all instances of trademark/copyright symbols throughout the text, including:

- TM on page: 20.

3) Some material included in your submission may be copyrighted. According to PLOSu2019s copyright policy, authors who use figures or other material (e.g., graphics, clipart, maps) from another author or copyright holder must demonstrate or obtain permission to publish this material under the Creative Commons Attribution 4.0 International (CC BY 4.0) License used by PLOS journals. Please closely review the details of PLOSu2019s copyright requirements here: PLOS Licenses and Copyright. If you need to request permissions from a copyright holder, you may use PLOS's Copyright Content Permission form.

Potential Copyright Issues:

i) Figure 2A. Please confirm whether you drew the images / clip-art within the figure panels by hand. If you did not draw the images, please provide (a) a link to the source of the images or icons and their license / terms of use; or (b) written permission from the copyright holder to publish the images or icons under our CC BY 4.0 license. Alternatively, you may replace the images with open source alternatives. See these open source resources you may use to replace images / clip-art:

ii) We note that [Figures 1A and 6] are created through BioRender. Please confirm that you hold a Premium account and provide a pdf copy of the CC BY 4.0 License as provided by BioRender. For instructions on how to generate a CC BY 4.0 license for your figure, please see the guidelines here: https://help.biorender.com/hc/en-gb/articles/21282341238045-Publishing-in-open-access-resources. 

If you are using the free assets from BioRender, we are unable to publish these images as they are licenced under a stricter licence than CC BY 4.0. In this case we ask you to remove the BioRender images and replace them with open source alternatives.

See these open source resources you may use to replace images / clip-art:

- https://bioart.niaid.nih.gov/

- https://bioicons.com/

- https://healthicons.org/

- https://scidraw.io/

- https://reactome.org/icon-lib

- https://www.phylopic.org/images

4) When completing the data availability statement of the submission form, you indicated that you will make your data available on acceptance. We strongly recommend all authors decide on a data sharing plan before acceptance, as the process can be lengthy and hold up publication timelines. Please note that, though access restrictions are acceptable now, your entire data will need to be made freely accessible if your manuscript is accepted for publication. This policy applies to all data except where public deposition would breach compliance with the protocol approved by your research ethics board. If you are unable to adhere to our open data policy, please kindly revise your statement to explain your reasoning and we will seek the editor's input on an exemption. Please be assured that, once you have provided your new statement, the assessment of your exemption will not hold up the peer review process.

**Reviewers' Comments:**

Reviewer's Responses to Questions

**Part I - Summary**

Reviewer #1: Lewis et al. assess the role of HIV-1 Vpr and HDAC inhibition in latent reservoir formation and HIV-1 promoter activity. They tested a series of proviral constructs in primary CD4+ T cells and determined that Vpr promotes LTR activity in the presence of the HDAC inhibitor Vorinostat. Further examination of this phenomenon suggested that Vpr promotes a central memory phenotype and that these cells are more resistant to Vpr-induced apoptosis. While several of the experimental observations herein confirm previous observations from prior studies, the novelty comes from the extensive use of primary CD4+ T cells. I have some concerns regarding the interpretation of some of the results and the magnitude of the changes induced by Vpr. Questions and concerns are detailed below.

Reviewer #2: Lewis et al explores activities of HIV-1 Vpr in CD4+ T cells with a focus of on whether Vpr influences HIV-1 transcription and the establishment of latency. Key findings are that Vpr and HDACs may interact to repress HIV transcription, and this activity is dependent on p300. In addition, Vpr influences primary CD4+ T cell survival and an expansion of Tcm cells, which have been shown to harbor latent proviruses. Strengths of the paper are the general approach of utilizing primary CD4+ T cells, the use of HIV-1 clones with a single accessory protein and utilizing Vpr mutations to map domains to specific functions that suggest Vpr may be through multiple mechanisms influencing HIV replication in primary CD4+ T cells. However, the potential multiple activities of Vpr, clouds the primary message of the paper which tries to emphasize the Vpr-HDAC-p300 axis. There are also some concerns as to the modest data and lack of exploring and discussing direct and indirect mechanisms of Vpr and HIV transcription. Specific comments are below.

The data in which Vpr+Vorinstat, a HDACi, examines how HDACs might alter HIV-1 infection, transcription, and latency show about a two-fold difference in HIV-1 expression, as monitored by GFP MFI and HIV-1 transcription. There is no discussion whether these differences are significant.

It is difficult to know if Vpr+Vor reflects difference in transitioning to latency since no experiments are performed to look at the ability to reactivate proviruses after the extended culturing.

It is stated that the percentage or number of cells that are expressing HIV do not significantly change, however, the flow cytometry profiles for the different experiments and different viruses suggest a great deal of variation between the different viruses and experiments. Does this reflect different fitness between viruses, experimental variation, or some other variable?

The use of different HDAC inhibitors would be interesting in that it might address potential off-targets of Vor, and provide insights into the role of different classes of HDACs.

The data with the trans expression of Vpr seemed modest and at times contradictory to the data obtained with the viruses. Are these significant differences?

The impact of Vpr on Tcm is interesting. It would be informative if they examined if this could lead to more latently infected cells by trying to reverse latency.

Has differential regulation of different HDACs been examined in T cell subset? This should at least be discussed in greater detail.

**Part II – Major Issues: Key Experiments Required for Acceptance**

Reviewer #1: - Why are the percent positive cells so different between the Thy1.2 and eGFP markers for the flow results? Based on the system, I would have anticipated that percent positive for eGFP would be lower, or at least equivalent, to percent Thy1.2 positive cells. For example, in Figure 1C 63% of the cells are eGFP positive while only 20% are Thy1.2 positive? This changes to 92% eGFP positive and 63% Thy1.2 positive after Vorinostat treatment.

- The bar graphs throughout are a little misleading for assessing the contribution of Vpr to reactivation of latent reservoirs with vorinostat. In general, Vpr expressing cells activate at ~10-fold lower frequency compared to control cells (ex., Figure 1C control is 18% whereas Vpr cells 1.13%; Figure 3C control is 15.1% while Vpr is 0.62%; Figure 4 control is %6.52 while Vpr is 0.23%). The bar graphs suggest those Vpr infected cells contain brighter eGFP expression, but the flow plots don’t seem to indicate that. Instead, the increase in MFI seems to be due to a smaller denominator, which may not be biologically relevant.

- The authors should validate that the increase in eGFP MFI in Vpr+vor cells translates to detectable increases in viral protein production compared to control+vor treated cells. This is particularly important given the discrepancy between the flow plots and bar graphs discussed above.

- It is odd that several of the Vpr mutants show significantly diminished protein abundance when expressed in cis compared to expression in trans, and that the eGFP MFIs were relatively equivalent. Did the authors confirm that the percent of cells that form latent reservoirs with these new viruses were relatively equivalent compared to wild-type?

- Because several previous studies have suggested that Vpr recruits p300 to the LTR to enhance transcription [PMIDs: 9560267, 12379213, 10505122], I would have anticipated that Vorinostat + p300 inhibition would block the increase in eGFP MFI. The data in the main figures in conjunction with the data presented in Figure S6 suggest this is not the case.

- Figure 4A indicates a strong increase in CD45A+-CCR7+ cells in Vpr infected cells compared to control cells, but there are significantly fewer Vpr infected cells in the CD45A+-CCR7+ flow plot. Could this increase be due to sampling bias? In other words, if the events were normalized between the plots would the increase observed in the Vpr samples be more comparable to control? For example, the total events in Figure 4B are more similar and the increase in CD45A+-CCR7+ cells in the Vpr infected cells is negligible compared to control cells.

Reviewer #2: No key experiments. Using different HDAC inhibitors would be informative.

Experiments that directly address latency would enhance the significance of findings.

In addition, demonstrating Vpr and p300 complexes at the LTR would strengthen the model.

**Part III – Minor Issues: Editorial and Data Presentation Modifications**

Reviewer #1: - I’m not sure if it an upload issue or a conversion issue but many of the figures are blurry and the legends are too difficult to read. This is particularly important for trying to interpret the flow cytometry data when comparing percentages across populations.

- Percentage label in Vpr trans mCh experiment in Figure 2D is offset to the right overlapping the other flow plot

- S5A labels are shifted

Reviewer #2: Minor comment: Potential indirect effects on HIV transcription should be included on the figure 6 model. Plus, I am not convinced their data support a mechanisms of Vpr+Vor directly influences LTR activity.

Minor comment: ref 21 and 23 are the same?

PLOS authors have the option to publish the peer review history of their article (what does this mean? ). If published, this will include your full peer review and any attached files.

**Do you want your identity to be public for this peer review?** For information about this choice, including consent withdrawal, please see our Privacy Policy .

Reviewer #1: No

Reviewer #2: No

**Figure resubmission:**

**Reproducibility:**



---

## [Decision Letter · Decision Letter 1]

15 Aug 2025

Dear Ms. Lewis,

We are pleased to inform you that your manuscript 'HIV-1 accessory protein Vpr possesses a cryptic p300-dependent transcription-promoting activity that is blocked by histone deacetylases in CD4+ T cells.' has been provisionally accepted for publication in PLOS Pathogens.

Best regards,

David T. Evans

Academic Editor

PLOS Pathogens

Richard Koup

Section Editor

PLOS Pathogens

Sumita Bhaduri-McIntosh

Editor-in-Chief

PLOS Pathogens

orcid.org/0000-0003-2946-9497

Michael Malim

Editor-in-Chief

PLOS Pathogens

orcid.org/0000-0002-7699-2064

Reviewer Comments (if any, and for reference):

Reviewer's Responses to Questions

**Part I - Summary**

Reviewer #1: The authors have adequately addressed my previous comments/concerns

Reviewer #2: The revised manuscript addressed most of my concerns including better statistical explanations, additional experiments that address mechanism, and expanded discussion.

**Part II – Major Issues: Key Experiments Required for Acceptance**

Reviewer #1: The authors have adequately addressed my previous comments/concerns

Reviewer #2: No major issues

**Part III – Minor Issues: Editorial and Data Presentation Modifications**

Reviewer #1: The authors have adequately addressed my previous comments/concerns

Reviewer #2: None

PLOS authors have the option to publish the peer review history of their article (what does this mean? ). If published, this will include your full peer review and any attached files.

**Do you want your identity to be public for this peer review?** For information about this choice, including consent withdrawal, please see our Privacy Policy .

Reviewer #1: No

Reviewer #2: No

---

## [Editor Report · Acceptance letter]

Dear Ms. Lewis,

We are delighted to inform you that your manuscript, " 

HIV-1 accessory protein Vpr possesses a cryptic p300-dependent transcription-promoting activity that is blocked by histone deacetylases in CD4+ T cells.," has been formally accepted for publication in PLOS Pathogens.

Best regards,

Sumita Bhaduri-McIntosh

Editor-in-Chief

PLOS Pathogens

orcid.org/0000-0003-2946-9497

Michael Malim

Editor-in-Chief

PLOS Pathogens

orcid.org/0000-0002-7699-2064